# You Are What You Eat: Application of Metabolomics Approaches to Advance Nutrition Research

**DOI:** 10.3390/foods10061249

**Published:** 2021-05-31

**Authors:** Abdul-Hamid M. Emwas, Nahla Al-Rifai, Kacper Szczepski, Shuruq Alsuhaymi, Saleh Rayyan, Hanan Almahasheer, Mariusz Jaremko, Lorraine Brennan, Joanna Izabela Lachowicz

**Affiliations:** 1Imaging and Characterization Core Lab, King Abdullah University of Science and Technology (KAUST), Thuwal 23955-6900, Saudi Arabia; abdelhamid.emwas@kaust.edu.sa; 2Environmental Technology Management (2005-2012), College for Women, Kuwait University, P.O. Box 5969, Safat 13060, Kuwait; nahla.alrifai@gmail.com; 3Biological and Environmental Sciences & Engineering Division (BESE), King Abdullah University of Science and Technology (KAUST), Thuwal 23955-6900, Saudi Arabia; kacper.szczepski@kaust.edu.sa (K.S.); shuruq.alsuhaymi@kaust.edu.sa (S.A.); mariusz.jaremko@kaust.edu.sa (M.J.); 4Chemistry Department, Birzeit University, Birzeit 627, Palestine; sarayyan@birzeit.edu; 5Department of Biology, College of Science, Imam Abdulrahman Bin Faisal University (IAU), Dammam 31441-1982, Saudi Arabia; halmahasheer@iau.edu.sa; 6Institute of Food and Health and Conway Institute, School of Agriculture & Food Science, Dublin 4, Ireland; lorraine.brennan@ucd.ie; 7Department of Medical Sciences and Public Health, University of Cagliari, Cittadella Universitaria, 09042 Monserrato, Italy

**Keywords:** metabolomics, NMR, MS, food, nutrition

## Abstract

A healthy condition is defined by complex human metabolic pathways that only function properly when fully satisfied by nutritional inputs. Poor nutritional intakes are associated with a number of metabolic diseases, such as diabetes, obesity, atherosclerosis, hypertension, and osteoporosis. In recent years, nutrition science has undergone an extraordinary transformation driven by the development of innovative software and analytical platforms. However, the complexity and variety of the chemical components present in different food types, and the diversity of interactions in the biochemical networks and biological systems, makes nutrition research a complicated field. Metabolomics science is an “-omic”, joining proteomics, transcriptomics, and genomics in affording a global understanding of biological systems. In this review, we present the main metabolomics approaches, and highlight the applications and the potential for metabolomics approaches in advancing nutritional food research.

## 1. Introduction

Society’s awareness of the importance of food quality arose at the beginning of the nineteenth century, when French lawyer Anthelme Brillat-Savarin wrote, in *Physiologie du Gout, ou Meditations de Gastronomie Transcendante*: “Dis-moi ce que tu manges, je te dirai ce que tu es.” (En. “Tell me what you eat and I will tell you what you are”]. The twentieth century, marked by successive wars, shifted food quantity to the fore, and humanity focused on mass food production, ignoring the efficiency and quality of nutritional products. Nowadays, we are experiencing a healthy-eating renaissance that has been unexpectedly further amplified by the COVID-19 pandemic [1].

A properly balanced diet, based on the qualitative and quantitative analysis of each product, can be used as supportive therapy in metabolic disorders, including those with a genetic etiology (e.g., a ketogenic diet for renal dysfunction [2]). Food products with particular functions (often related to health promotion or disease prevention), made by adding new ingredients or more of the existing ingredients, are termed “functional food”. The global market for the functional food industry (food, beverage, and supplements) has been growing extremely fast in recent years, reaching USD 177,770.00 million in 2019 [3]. In order to increase the importance of nutrition in the global market, new techniques associated with global data-analysis systems are needed for food analysis.

Metabolomics qualitatively and quantitatively defines metabolites (small molecules) present in biological samples, and is becoming increasingly important in nutritional research (Figure 1). Metabolite screening provides a wealth of biological data, such as diet intake, drug administration molecular fingerprints, and a snapshot of metabolism. Additionally, metabolomics approaches enable monitoring of metabolites in correlation with genetic and environmental components, including age, gender, drug toxicology, lifestyle, health status and most notably nutrition intake [4,5,6,7,8,9]. Different metabolomics approaches are employed in nutrition research of food components, body fluids, and biological tissues analysis. Moreover, physiological responses to a particular food regimen [10,11,12,13,14] are studied with metabolomics. In this review, we will discuss the NMR technique while it is widely used in the metabolomic profiling of selected food products, which significantly influence the global food market. Additionally, we will present the largest databases gathering metabolomics data of food products.

## 2. (Un)Targeted Metabolomics

Metabolomics is “the measurement of metabolite concentrations and fluxes and secretion in cells and tissues, in which there is a direct connection between the genetic activity, protein activity and the metabolic activity itself” [15], while metabonomics is “the quantitative measurement of the multivariate metabolic responses of multicellular systems to pathophysiological stimuli or genetic modification” [15,16]. Nowadays, metabolomics and metabonomics are used to describe similar research approaches, and in this review the term metabolomics will be used even when referring to papers that used metabonomics terminology.

Metabolomics analysis can be either targeted (i.e., focused on quantitative measurements of usually small numbers of metabolites) or untargeted (i.e., focused on metabolic profiling of the total complement of metabolites for the studied samples) [17,18]. Untargeted analysis screens the metabolites with the intention to compare a profile or a pattern among different sub-groups of samples. The untargeted analysis is frequently used in nutrition research to profile food molecular composition, determine the outcome of dietary intervention on human metabolism, and to characterize an individual’s metabolic phenotypes. The untargeted analysis usually starts with metabolite extraction from studied samples, where different extraction methods can be utilized sequentially to maximize the extracted molecules. Untargeted analysis is frequently combined with statistical analysis (such as multivariate), and pattern recognition analyses, to perceive changes in metabolites and within sub-groups of samples. In addition to valuable information that can be extracted from statistical approaches, additional data from metabolomics database(s) and/or using standard samples are required for evaluating the outcome, such as reporting new metabolites.

Conversely, targeted metabolomics focuses on analysis of a number of selected metabolites, such as studies related to specific metabolic pathways, drug toxicology, and specific effects of certain foods/diets [19,20,21]. Targeted metabolomics approaches are mostly based on hypothesis-driven investigation, where specific quantification methods are optimized to quantify the concentration of the specific metabolites. In nutrition research, the targeted metabolomics approaches are used in different aspects, for instance to determine the impact of certain diets/foods on metabolic pathways, and dietary effects on human health.

## 3. Metabolomics Analytical Tools

Metabolomics employs different analytical platforms including high-performance liquid chromatography (HPLC) [22,23,24,25,26,27], Fourier transformed infrared (FT-IR) spectroscopy [28,29,30,31], mass spectrometry (MS) [32,33,34,35] and nuclear magnetic resonance (NMR) spectroscopy [36,37,38,39,40,41,42].

Next to the conventional HPLC [35,43,44], ultra-performance LC (UPLC) has been developed, which utilizes smaller beads in the column (less than 2 um) and operates under higher pressure than conventional LC, offering a significant reduction of band broadening. In this way, it increases sensitivity up to 2–3 times over the standard HPLC [44,45]. Moreover, UPLC is a cost-reducing method that requires a smaller sample and a shorter time for measurement [44,45].

MS is a sensitive analytical tool, capable of detecting metabolites at very low concentrations, but it requires a preliminary separation step, such as gas chromatography or liquid chromatography [46].

The minimal sample preparation requirements, high reproducibility, and non-destructive character are major advantages of the NMR analytical technique in metabolomics analysis. Another important advantage of the NMR technique is that it can be used both for identification and quantification of the analyzed molecules although qualitative analysis of a multi-component sample is complicated. In addition, the NMR signal’s low sensitivity and overlap can be improved using recent technological advances (hardware, pulse sequence and spectral acquisition) [47]. Although low sensitivity is the main limitation of NMR spectroscopy, significant developments have been made to enhance the sensitivity, including micro probes [48], cryogenically-cooled probes [49] and the dynamic nuclear polarization (DNP) approach [50,51,52].

Nowadays, it is not possible to analyze all metabolites with one analytical tool due to molecule diversity with respect to size, polarity, concentration levels and stability. Metabolomics data can be enriched by a sequential combination of different analytical techniques. For instance, separation signals can be improved with NMR-coupled gas chromatography, (GC-NMR), liquid chromatography (LC-NMR) and solid phase extraction (LC-SPE-NMR). Nevertheless, few laboratories have a facility of combined techniques due to high costs and in practice, combine techniques are used rarely. In the coming years NMR-based analysis will most likely be favored by scientists as the essential analytical approach that provides reproducible results [53,54,55].

A wide range of NMR experiments are available and commonly used in metabolomics applications [42,56,57,58,59,60,61,62,63] and later on we will focus on NMR results. For instance, one-dimensional (1D) experiments are used for detection and quantification of compounds containing atoms such as ^1^H, ^13^C, ^15^N, ^31^P [37,64]. More sophisticated experiments such as two-dimensional (2D) correlation spectroscopy (COSY), total correlation spectroscopy (TOCSY), and heteronuclear single quantum correlation spectroscopy (HSQC) are usually used to remove the overlapping effects of peaks on NMR spectra, and to allow the detection and identification of more metabolites than with 1D NMR [37,42,65,66,67,68,69].

Selected examples of NMR data in nutritional metabolomics studies are presented in Table 1.

## 4. Foodomics Databases

Foodomics is defined as “a discipline that studies the Food and Nutrition domains through the application and integration of advanced -omics technologies to improve consumers’ well-being, health, and knowledge” [90].

The metabolomics workflow starts with a biological sample preparation for analysis (Figure 2), which needs to satisfy standards for the chosen analytical method. The numerous analytical data are analyzed with statistical methods and then gathered in the databases.

Although foodomics is a fairly new discipline, there is an abundance of databases related to the topic. Those databases provide various information about the product of interest—from NMR/MS reference spectra of a product’s metabolites, to the cDNA clone libraries, full-length mRNA sequences, gene structures and expression profiles of genes. That information can be later used to further identify/analyze products of interest, whether metabolites or genes.

One example is TOMATOMICS (http://plantomics.mind.meiji.ac.jp/tomatomics/, accessed on 13 January 2021)—a web database of genetic information of the *Solanum lycopersicum* (tomato) [91]. The TOMATOMICS database provides users with Basic Local Alignment Search Tool (BLAST) search functions and a JBrowse-based genome browser. The BLAST program is used to align query sequences with those stored within the database in order to find sequence homology, and provides further information about the significance of each alignment [92,93]. On the other hand, the JBrowse-based genome browser allows the visualization and comparison of genome annotations, transcripts, variations, and T-DNA insertion sites between different cultivars [91,94]. Additionally, TOMATOMICS provides information on locus groups and visualizes them using the JBrowse browser. However, TOMATOMICS is not the only database that focuses on tomatoes. TOMATOMA (https://tomatoma.nbrp.jp/, accessed on 13 January 2021) collects information about mutant lines of the Micro-Tom cultivar generated by ethyl methane sulfonate (EMS) treatment or γ-ray irradiation [95,96]. What differentiates TOMATOMA from TOMATOMICS is the fact that TOMATOMA offers a set of metabolic information, such as Brix values and carotenoid content values, of the mutant fruits. Both of these factors affect the preferences of consumers as Brix values provide information about the sweetness of the fruit (therefore the taste) while carotenoids act as an indicator for nutritional value. This database can also provide users with Micro-Tom mutant seeds that can be used for further investigation [95].

Another database related to foodomics is the Tea Metabolome database (TMDB) [97]. This database provides users with information about small chemical compounds found in *Camellia* spp., with a special focus on *Camellia sinensis*. TMDB, ^1^H NMR and ^13^C NMR spectra with similarity identification as well as MS and MS/MS data can be obtained for the purpose of identification of metabolites. As for the amount of records, at the time of creation (2014), it contained more than 1473 compound entries with more than 30,000 different data entry fields [97]. The entries presented in TMDB were collected based on the information obtained in 364 published books, journal articles, and electronic databases. In 2014, the number of compounds recorded for different types of tea was presented: 713 compounds in green tea, 497 in black tea, 140 in oolong tea, and 445 for dark tea. Most of the compounds (74%) collected in the database have a molecular weight of less than 500 Da. Additionally, each compound entry in TMDB has a bioactivities data field, which describes the biochemical effects on the cells [97].

Databases that focus on meat and dairy products also exist, such as the Bovine Metabolome Database (BMDB) (www.bovinedb.ca, accessed on 13 January 2021) [98]. This database contains in total 51,801 metabolites, and only a small fraction (4.1%) of metabolites with unique structures have been quantified [98]. Each of the metabolites stored within the database have their own structures in multiple formats, basic descriptions, chemical ontology, physico-chemical properties, their reference spectra (NMR, GC-MS, and LC-MS), pathway information, and literature citations from the scientific literature [98]. The metabolites in BMDB have been linked to eight bovine tissues and six different bovine biofluids. For the tissues, the most information about the metabolites can be found for the liver, with 1254 identified metabolites with unique structures, of which only 273 were quantified [98]. As for biofluids, milk has been the most widely investigated, with 2350 unique metabolites identified, with around 70% of them (1652) being quantified [98]. As a complex biofluid, milk itself has a dedicated database called the Milk Composition Database or MCDB (http://www.mcdb.ca/, accessed on 13 January 2021) storing more than 19,000 spectra (NMR, LC-MS, GC-MS) [99].

In addition to the aforementioned databases, a group of more general ones exist that focus not only on one product or products from the same origin but on diverse types of food. For example, FooDB (https://foodb.ca/, accessed on 13 January 2021) is the world’s largest open-access database that provides information on macronutrients and micronutrients as well as compound nomenclature, descriptions, information on structure, chemical class, physico-chemical data, food source(s), color, aroma, taste, physiological effects, presumptive health effects (from published studies), and concentrations in various foods [100]. Currently, almost 800 types of products are listed, with more than 70,000 compounds characterized [100]. In addition, databases that have a particular focus on a group of compounds, rather than types of food, also exist. A prominent example is PhytoHUB (http://phytohub.eu/, accessed on 13 January 2021), which specializes in phytochemicals and their human and animal metabolites [101]. The content of PhytoHUB consists of about 1850 entries of which around 1200 are polyphenols, terpenoids, alkaloids, and other plant secondary metabolites, with 560 human or animal metabolites [101]. The total number of plant-based foods featured in PhytoHUB, as of 2021, is 371 [101].

## 5. Foodomics as a Standard for Safety and Quality Assessment

The quality of food is one of the most important factors that is considered when choosing products [102,103,104,105]. The preference for food quality over quantity and price have been extensively observed among individuals with a higher income and higher level of education [103,106]. Since global education levels across many areas are rising [107], and a link between level of education and health awareness has been observed [108,109], one can expect a constant increase in the preference for food quality throughout the next decades [103,106]. This will require utilization of new, reliable, and rapid methods to meet demand. Luckily, recent progress in the field of foodomics has demonstrated the efficiency of NMR and MS methods to resolve problems related to safety evaluation and establishing standards of quality on different products.

A recent example is the MEATabolomics approach that focused on the identification of potential biomarkers to control meat quality and safety [110]. Assessment of meat quality from the consumers’ perspective mainly involves appearance (including the color) [110,111,112]. The red color of meat is mainly due to the presence of myoglobin, although factors such as the structure of the tissue, pH, muscle source, presence of antioxidants or lipid oxidation also play significant roles [113,114]. In this case, gaining insight into the metabolomics effects on the color of meat, as well as the environmental factors affecting their changes, are crucial factors for increasing physical quality.

One study related to the biochemical changes and their impact on meat color was undertaken by Ramanathan et al. [115]. Their goal was to determine the differences in metabolite profile and mitochondrial content between normal-pH and dark-cutting beef. For that, a GC-MS spectrometer-based nontargeted metabolomic approach was taken that revealed the downregulation of glycolytic metabolites and the upregulation of tricarboxylic substrates in dark-cutting beef when compared to normal-pH beef. Additionally, dark-cutting beef had greater mitochondrial content and higher levels of neurotransmitters such as 4-aminobutryic acid and succinate semialdehyde. The authors speculated that the mitochondrial content and downregulation of metabolites involved in glycolytic pathways resulting in lower lactic acid formation during anaerobic metabolism would result in a high-pH of dark-cutting beef [115].

Another study by Beauclercq et al. [116] characterized the metabolomics signature of muscle and serum of two chicken lines with almost 17% difference in glycogen content in the breast muscle. The differences were investigated by quantification of muscle metabolites by high-resolution NMR (^1^H and ^31^P) and serum metabolites by ^1^H NMR. The analysis identified 20 and 26 discriminating metabolites in muscle and serum, respectively, between the two lines, the majority of which were related to carbohydrate metabolism and the production of energy such as glycogen or glucose 6-phosphate. The main conclusion however was that the pHu-line (with a low ultimate pH of the meat) used carbohydrate as the main source of energy, whereas those in the pHu+ (high pH of the meat) used energy produced from amino acid catabolism and lipid oxidation, which leads to an adaptive response to oxidative stress. Additionally, 15 biomarkers were identified that could help to distinguish poultry likely to produce meat with high or low pHu values, thereby improving the quality of sold meat [116].

## 6. The Impact of Food Metabolites on Human Health

The changes in metabolite levels in a person are affected by many factors, including genetics, the environment, and dietary intake [117]. The type of diet is not only directly responsible for the metabolites but also indirectly via nutritional changes inflicted upon the gut microbial and thereby their own metabolism [117,118]. In recent years, more evidence has surfaced proving that many aspects of human health are affected by the gut microbiota such as obesity-associated disorders, regulation of blood pressure, and the immunity of the host to pathogens [119,120,121]. There are approximately 10^14^ bacterial cells in the gastrointestinal tract of a human, which is about 10 times the total number of human cells in the body, with a total biomass of 2 kg [122,123]. These bacteria belong to more than 1000 different species [122], some of which are crucial for our wellbeing, such as lactic acid bacteria and Bifidobacteria for their ability to synthesize vitamin K as well as many B vitamins, such as biotin, cobalamin, folates, nicotinic acid, pantothenic acid, pyridoxine, riboflavin, and thiamine [124,125]. Furthermore, the functionality of gut microbiota is modified by their environments in response to dietary changes, thereby improving human dietary flexibility. The microbiota and its metabolic pathways are influenced by the genetics, geographical regions, diets, antibiotics and other therapies of the host [126].

In recent work, Farag et al. [123] evaluated the impact of functional food, which is defined as a dietary supplement that beneficially regulates body functions, on the microbiota with a focus on the microbiota physiology that could be evaluated by untargeted metabolomics. Seven functional foods, such as green tea, black tea, *Opuntia ficus-indica* (prickly pear, cactus pear), black coffee, green coffee, pomegranate, and sumac were introduced to consortium culture with eight different types of bacteria: *Anaerostipes caccae*, *Bacteroides thetaiotaomicron*, *Bifidobacterium longum*, *Blautia producta*, *Clostridium butyricum*, *Clostridium ramosum*, *Escherichia coli* and *Lactobacillus plantarum* that would resemble metabolic activities found in the human gut [123]. Samples of functional food extract and from blank culture were analyzed using GC coupled to MS detection (GC-MS). The results stated that 131 metabolites were identified, including organic, amino, fatty, nucleic acids, alcohols, sugars, inorganic, nitrogenous compounds, phenolics, and steroids. The most abundant class in the cultures was amino acids. Moreover, multivariate data analysis was performed to identify samples and examine how functional foods influence gut microbiota metabolisms. These results provided insights into how functional foods modulate gut-microbiota metabolism by either inducing or inhibiting specific metabolic pathways such as increased GABA production in the presence of higher acidity induced by metabolites such as polyphenols and organic acids, or purine alkaloids such as caffeine acting as precursors of purine by microbiota demethylation [123].

Another interesting aspect of metabolomics is its ability to obtain information on an individual’s diet from food-induced shifts in metabolites.

Analysis of dietary patterns allows researchers to gain a broader insight into dietary intake and applying metabolomics to achieve this is promising. Untargeted metabolomic profiles were employed to distinguish between two Nordic dietary patterns used in an intervention study; the New Nordic Diet (NND) or the Average Danish Diet (ADD) [127]. Using the metabolomic data a multivariate model was established, which classified the two dietary patterns with a low misclassification error rate (19%). A study by Posma and colleagues [128] highlighted the power of such metabolomics approaches. Urine was collected from 1848 Americans, and ^1^H NMR spectroscopy was used to measure the urinary metabolome, producing a wide range of chemical profiles. It was observed that 46 metabolites can differentiate between people with healthy and unhealthy dietary patterns, such as an association between sodium and calcium with citrate and formate on blood pressure, adiposity, and renal function, and a correlation between fructose, glucose, and vitamin C with biomarkers of citrus fruit consumption such as prolinebetaine, 4-hydroxyprolinebetaine and 2-hydroxy-2-(4-methylcyclohex-3-en-1-yl)propoxy glucuronide [128].

## 7. NMR in Analysis of Food Components

In general, food and drink component analysis include the identification, quantification, and classification of food constituents such as carbohydrates, lipids, hormones, nucleic acids, vitamins, and minerals. Metabolomics allows the detection and characterization of hundreds of biological sample components simultaneously, offering powerful tools for more comprehensive and detailed pictures of food composition. This provides a means to analyze food components for certain important pursuits, such as knowing the bio-active molecules, food quality, and authenticity, as well as searching for significant nutrients. For example, Kim et al. [80] used ^1^H NMR coupled with multivariate statistical analysis to monitor changes in metabolic profiling of raspberry fruits at different ripening stages using multiple extraction and NMR dissolution solvent condition systems. NMR-based metabolomics approaches were also used to analyze the polar portion of methanol extracts of celery seeds, and several compounds were identified, including sesquiterpenoid glucosides, norcarotenoid glucosides and phthalide glycosides, and their structures were determined [129].

In the next section, we will highlight several examples of profiling food constituents using NMR-based metabolomics approaches.

### 7.1. Tomato

Perez et al. [130] performed a metabolomic analysis of tomato fruit and tomato tissues at three different ripening stages using high resolution magic angle spinning (HRMAS) NMR. NMR data of whole fruits were obtained, which showed a spectral resolution like that of solution ^1^H-NMR with the advantages of minimal sample manipulation and the possibility of simultaneous analysis of polar and non-polar metabolites. The same technique was used by Perez et al. [131] to investigate differences at the metabolic level between flavor varieties of high quality tomatoes from Spain. The results showed clear differences between varieties as a function of the ripening process, and revealed the existence of variety-dependent relationships between external appearance and metabolic content.

Stark et al. [132] reported a comprehensive ^13^C NMR relaxation study of hydrated tomato fruit cuticle that discovered changes in the individual polymer motion of the cutin/wax components of the tomato cuticle, and the interaction of these components within intact cuticles in response to the addition of water. Additionally, they reported the effect of thermal stress on biomacromolecular dynamics on different timescales. The influence of these factors on protective plant covering characteristics was also discussed.

Le Gall et al. [82] showed that NMR combined with chemometrics and univariate statistics can successfully trace differences in metabolite levels between genetically modified tomatoes and non-modified (control) tomato plants, grown side by side under the same conditions, and detected potential unintended effects in the genetically modified crops.

De Falco et al. [133] analyzed the changes in metabolites of three different tomato genotypes (tolerant “T”, susceptible “S”, and “F1” hybrid obtained between T and S) after exposure to *Tuta absoluta* using ^1^H-NMR. *T. absoluta* “tomato leafminer, tomato moth” is considered to be one of the most harmful tomato crop pests causing yield loss. NMR-based metabolomics approaches coupled with multivariate data analysis were performed to profile detailed metabolites of the control and the pest exposed samples on the three different tomato genotypes (T, F1, and S), which could explain the chemical diversity of the signaling compounds that contribute to defense mechanisms in the plants. Results showed that γ-aminobutyric acid (GABA) signals were much higher in all the samples that were exposed to *T. absoluta* compared to the control tomato samples. Moreover, organic acids such as fatty acids and acyl sugars, chlorogenic acid, neo-chlorogenic acid, and feruloyl quinic acid were much higher in the tolerant (T) genotype exposed samples, suggesting the correlation with exposure to leafminer. They also showed that trigonelline increased in all tomato varieties after exposure to *T. absoluta*. The authors concluded that metabolomics analysis may provide fundamental insights for better understanding of the tomato–pest interactions and for enhancing tomato breeding in agriculture [133].

In another study, Meza et al. [134] discovered that tomato quality was improved by growing under moderate salt stress, which may allow for sustainable fruit yields. It is known that salinity can affect fruit quality by changing its metabolic pathways, which depends on the fruit variety and salt concentration. Two traditional tomato varieties grown in the Mediterranean region were chosen because of their genetic diversity that may give a comprehensive view of fruit quality traits. Planting the two varieties with no salt stress as a control, and with moderate salt stress (50 mM NaCl), had no effect on fruit yield of either variety. Fruit quality traits, including primary and secondary metabolites, were analyzed in those two Mediterranean tomatoes (Tomate Pimiento “TP” and Muchamiel Aperado “MA”). ^1^H- NMR was used for the analysis of the primary metabolites, and UHPLC to analyze the secondary metabolites. Results showed that primary metabolism for both tomato varieties was similar, and in “TP” fruits, the highest constituents were three free amino acids present in tomato, GABA, glutamate, and glutamines, which were increased also under salinity. The most important shift in secondary metabolites attributed to salinity was α-tocopherol, which increased in both “TP” and “MA” red mature fruit. Meza et al. [134] concluded that the two tomato varieties can be considered good sources for genetic diversity in breeding because of their ability to improve the quality of tomato fruit under moderate salt stress.

### 7.2. Green Tea

*Camellia sinensis* L. is popularly known as green tea, and is one of the most consumed beverages in the world. There are many primary metabolites, such as organic acids, amino acids, and carbohydrates, present at different concentrations in the tea leaves. In addition, secondary metabolites, which include alkaloids and polyphenols, are also present that influence many biological and pharmacological activities, such as anti-microbial, anti-tumor, and anti-oxidative properties. In addition, the quality of the tea leaves is determined by these metabolites, which are influenced by different factors such as species type, geographical status, and climate factors.

There are several metabolomics studies that have focused on profiling the chemical composition of green tea [135,136,137,138]. Lee et al. studied the changes in green tea metabolites during tea fermentation using ^1^H NMR coupled with multivariate statistical analysis, and were able to distinguish between metabolic profiling of green tea and fermented tea [139]. NMR-based metabolomics approaches were used to study the effects of geographical origin, climate, cultivar, and manufacturing and cultural practices on tea chemical compositions by analyzing 180 tea samples collected from Japan, South Korea, and China [136]. The results showed a clear correlation between environmental factors and the metabolome of green, white, and oolong teas from all three countries [136].

Wahyuni et al. [140] identified the primary and secondary metabolites of three varieties of dried green tea leaves, which were grown at Kemuning, Indonesia, using ^1^H-NMR with two-dimensional NMR techniques such as J-resolved and 1H-1H COSY. Results showed that amino/organic acids and phenolic metabolites were detected in the spectra of the three varieties of green tea leaves, containing amino acid characteristics, theanine, and phenolic characteristics, and epicatechin derivates.

### 7.3. Olive Oil

The olive tree (*Olea europaea* L.) is widely cultivated in many parts of the world especially in Mediterranean countries [141]. Its cultivation date has been estimated to be 5000 B.C. One of the oldest olive trees in the world exists in Elwalaja, Bethlehem, Palestine, which was estimated to be more than 4000 years old (https://www.palestine-studies.org/en/node/78472, accessed on 13 January 2021). This tree, and similar ancient trees, have literally survived attacks from hundreds of generations of pathogenic organisms, indicating that they have developed a particularly efficient molecular defense system against pathogens [142]. Nowadays, in Mediterranean countries, there are more than 2000 varieties of olive tree [143]. The climatic conditions in these countries, like warm weather and long days of sunlight irradiation positively contribute to the cultivation of the olive trees and activate the synthesis of phenolic compounds in the fruits and leaves [144].

The quality of olives and olive oil has received much worldwide attention. To prevent fraud issues and ensure quality, the International Olive Council (IOC) issued guidelines for their sensory evaluation. Beteinakis et al. [145] developed a method to assess the quality of edible olives from the Konservolia, Kalamon and Chalkidikis cultivars collected from different areas of Greece and processed using the Spanish or the Greek method. They used so-called statistical total correlation spectroscopy (STOCSY) to measure specific biomarkers such as tyrosol, hydroxytyrosοl, verbascoside, luteolin, quercetin, maslinic acid, oleanolic acid, succinic acid, lactic acid, propionic acid, acetic acid, formic acid, triacylglycerols, linoleic acid and glycerol to assess the quality of the olives. Their findings suggest that the phenolic composition of the olives, at least for the Konservolia variety, is significantly influenced by the geographical origin, and more than by the cultivar. Additionally some cultivars were richer in some of the biomarkers than others, and the method of preparation also had an impact on the concentration of some biomarkers. They concluded that STOCSY revealed certain peak correlations, which, with input from the literature, led to an unprecedented peak assignment for these compounds in the total sample extracts. In addition, they also highlighted the value of STOCSY for supporting the assignment of biomarkers, which has been a limitation of untargeted metabolomics approaches [145].

The health benefits of olive oil have been attributed to several of its contents, specifically polyunsaturated fatty acids, polyphenols, and squalene, as well as oleocanthal, which is a secoiridoid phenolic compound with potential therapeutic properties against inflammation, cancer, and neurodegenerative diseases. The quality of olive oil has been the focus of many studies in the last few decades. Merchak et al. used NMR spectroscopy to study olive oils from different regions in Lebanon [146]. A total of 187 samples were tested using ^1^H NMR spectroscopy as a rapid metabolomic tool. Several illustrative variables related to olive oil contents such as fatty acids were measured and used in multivariate analyses. The samples were classified according to the geographical region, shape, size, and color of the olives, ripening period and latitude. Their results indicated some variations in the composition of oil contents according to these classifications [146].

Similarly, Rongai et al. used NMR spectroscopy coupled with multivariate analysis to investigate the metabolomic profiles of more than 200 extra virgin olive oil (EVOO) samples from different countries compared with EVOOs of different regions in Italy [147]. Relatively high polyunsaturated fatty acid content, such as linolenic acid, was observed in Tunisian oil, while a high relative content of monounsaturated fatty acids was associated with all other oils. This result was partly attributed to the different rainfalls and temperatures between these countries. The authors concluded that simple climate predictors were not enough to identify correlations with specific EVOO metabolic profiles, and further studies of more detailed climate parameters were needed [147].

A second study by Rongai et al. used ^13^C NMR and multivariate analysis to discriminate between olive oil samples from different regions in Italy [148]. Their results indicated similarities between EVOOs from regions with similar geographical conditions, and some differences between EVOOs from regions with different geographical conditions, in agreement with the previous studies [148].

In conclusion, NMR spectroscopy can be used to distinguish and evaluate EVOOs, which helps to prevent instances of fraud and ensure quality.

## 8. Metabolomics in Food Quality Control

Food composition defines its quality and authenticity. Food authentication and quality control is a major concern in all countries and markets, where the declaration label should match the chemical composition and the origin of the food item. Metabolic fingerprinting has proven to be a powerful, rapid, and cost-efficient approach for evaluating food authentication and quality. Although metabolomics approaches have been used in the authentication and quality control studies of several foods, the applications of figure printing are still in the initial stages and no country has yet approved metabolomics for food validation. Thus, more studies with larger number of samples and databases are required. In particular, more studies should be conducted to investigate factors that may influence the fingerprinting features, such as climate factors, agricultural production systems, food processing, production systems, and genetics [127].

One of the earlier studies, from Roberto Consonni and Laura R. Cagliani, investigated the ability to distinguish polyfloral and acacia honeys based on their geographical point of origin (i.e., from the European Community or not), and potential differences in sugar isoforms [149]. They analyzed 41 honey samples (polyfloral and acacia) from different countries by ^1^H NMR and ^13^C spectroscopy, coupled with a multivariate statistical method (i.e., principal component analysis (PCA) and hierarchical projection to latent structures discriminant analysis (partial least-squares discriminant analysis (PLS-DA)). Their results showed a significantly higher fructose and sucrose content in acacia honey than in polyfloral honey. Additionally, a different ratio was observed between fructose forms βFP and βFF only for polyfloral honey with an Argentinean point of origin, while Hungarian samples showed resonance shifts for fructose forms of αFF, βFP, βFF, and the glucose form of αGP isoforms for both varieties. These data could be used as geographical markers for Argentinean and Hungarian honey [149].

Another interesting example is the study by Alonso-Salces et al. who studied fingerprints of virgin olive oil from Italy, Spain, Greece, France, Turkey, Cyprus, and Syria, from three different periods of harvests [150]. The researchers utilized ^1^H NMR and combined it with PCA, linear discriminant analysis (LDA), and PLS-DA, in order to determine geographical origin with the support of isotope ratio mass spectrometry (IRMS) of carbon (δ13C) and hydrogen (δ2H). Their results enabled the authors to create models that would help to distinguish geographical origin between Spain, Italy, and Greece. For example, the model distinguished virgin olive oil from Greece from the rest of the virgin olive oils with >97% success rate. For Italy and Spain, the models presented classification abilities of 89% [150].

In Table 2, we present a few more examples of recent studies where metabolomics approaches have been used to investigate food authentication.

## 9. Food Intake Promoting a Healthy Lifestyle

Intake of a Western diet (WD), which is rich in saturated fats and simple sugars, is associated not only with weight gain, diabetes, and metabolic diseases, but also with impaired hippocampal-dependent memory and hippocampal pathologies [155]. In contrast, a high content of ketone bodies may protect against the adverse effects of high adiposity and/or high blood glucose levels on hippocampal-dependent cognition [156].

The ketogenic (or keto) diet (KD) was elaborated in the 1920s for the treatment of refractory epilepsy. It produces similar physiological effects to that of fasting, and reduces the occurrence of epileptic seizures. In the 90s, the keto diet was extended to overweight patients, and patients with diabetes, metabolic syndrome, cancers, and specific psychiatric and neurological disorders [157], providing health benefits.

The nervous system appears to be particularly susceptible to dietary treatment. Recently, therapeutic diets for the management of autism spectrum disorder were reviewed [158]. Interestingly, the neuroprotective effect of a keto diet in animals might be modulated by the gut microbiota [159]. Moreover, the keto diet has been associated with a range of shifts in the gut microbiota, namely it influences microbial diversity, and reduces concentrations of saccharolytic taxa, including beneficial bacteria such as *Bifidobacteria spp* [157]. The keto diet excludes or limits major food groups (e.g., grains, dairy, and certain fruit and vegetables). In order to contrast the negative effects of the keto diet, vitamins with trace elements supplementation [160], and pre- or probiotics have been proposed [161,162].

Numerous therapeutic diets have become an integral part of the clinical treatment for obesity, dyslipidemias, diabetes, cardiovascular disease, and hypertension [163]. Integrative and functional medical nutrition therapy (IFMNT) is defined by in-depth assessment of a patient’s nutritional status followed by the implementation of a personalized therapeutic diet using food and targeted supplementation. The history and principles of therapeutic diets are reviewed elsewhere [164].

Metabolomics plays an important role not only in the analysis of nutrients in single food products, but also in the analysis of diet outcomes. A recent study with the use of a mouse model by Licha et al. presented biochemical reactions and pathways of KD in combination with chemotherapy [165]. Jin et al. reviewed how the metabolomics approach is use to investigate the effects of a Mediterranean diet, and the role of the microbiome [166]. Metabolomics is allowing researchers to understand mechanisms following dietary interventions, which in turn advance our knowledge of the relationships between diet and health/disease.

## 10. Conclusions and Future Perspectives

Metabolomics analytical approaches are forming the foundations of innovative nutritional science that is opening up new therapeutic opportunities and markets. Chromatography and NMR spectroscopy, separately or combined, deliver qualitative and quantitative data on the molecular contents of all food types, which can be collected and disseminated by databases. Food metabolomics data can be combined with metabolomics of life processes, and together depicting in detail the living matter at previously unknown levels. At this point, we have the capabilities to re-design the food market from strictly nutritional to therapeutic, and to be in equilibria with the environment.

## Figures and Tables

**Figure 1 foods-10-01249-f001:**
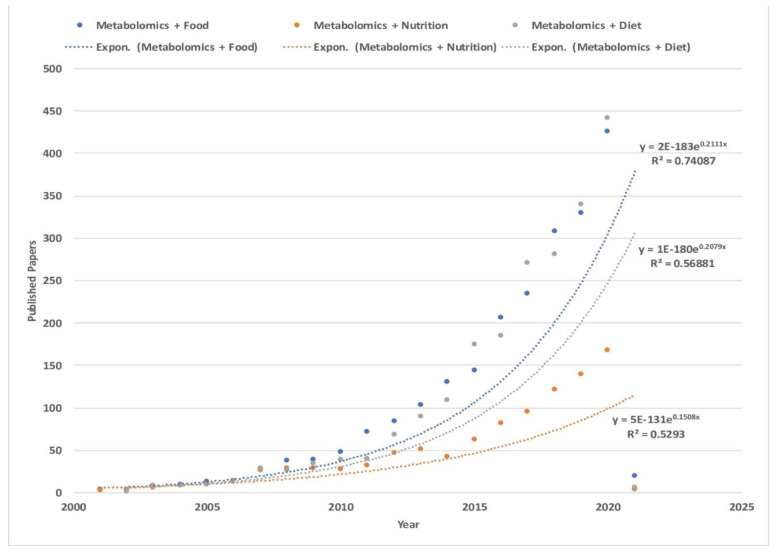
Number of published papers over the years. Each line represents search results conducted (13 January 2021) on Web of Science (www.webofknowledge.com, accessed on 13 January 2021) for: (blue) topic: (metabolomics) and topic: (food), (orange) topic: (metabolomics) and topic: (nutrition), and (gray) for topic: (metabolomics) and topic: (diet). Timespan: all years. Indexes: SCI-EXPANDED, SSCI, A&HCI, CPCI-S, CPCI-SSH, ESCI, CCR-EXPANDED, IC. The higher results of the search topics among metabolomic papers were for “metabolomics with food” with 2253 papers, followed by the topic “metabolomics with diet” with 2168 papers, and finally “metabolomics with nutrition” with 995 papers. Moreover, the topic “metabolomics with food” has shown an exponential increase over the years with R^2^ = 7, higher than both “metabolomics with diet and nutrition” with an exponential trend of R^2^ = 5, pointing to a positive exponential growth of interest among scientists towards the exploration of food metabolomics. Further information about the published papers in the three topics is in the Appendix A.

**Figure 2 foods-10-01249-f002:**
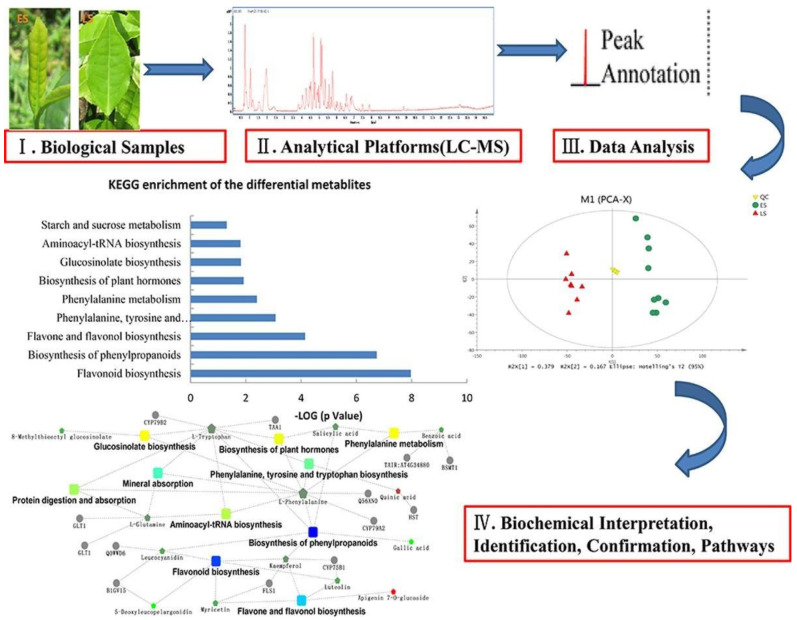
Schematic representation of metabolomics workflow in nutritional research.

**Table 1 foods-10-01249-t001:** Selected examples of metabolomic studies using NMR spectroscopy in food science.

Analytical Method	Example of Use in Sample Analysis	Purpose of Study	Reference
	Honey	Determination of the geographical origin of honey	[70]
	Watermelon	Description of metabolites in red and yellow watermelon cultivars with focus on the carotenoid group	[71]
	Pine mushrooms	Ilustration of differences among the various grades of pine mushroom	[72]
	Wine	Characterization of the metabolites in wines vinified in different continental areas and from different grape varieties to improve the quality of wine	[73,74]
	Green tea	Developing a method to evaluate the quality of Japanese green tea	[75]
	Ginseng (Panax ginseng)	Developing a method to control the quality of Ginseng commercial products	[76]
NMR	Fish tissue	Demonstrating the efficacy of NMR in environmental research/Optimization of tissue extraction methods	[77,78]
	Maize (Zea mays)	Safety evaluation of genetically modified maize	[79]
	Black raspberry fruit	Monitoring biochemical changes of black raspberry fruits based on the maturation level, extraction method and NMR-solvent conditions	[80]
	Tomato dry-powder with organic solvents	Profile and level characterization of carotenoids and flavonoid glycosides/Detection of potential unintended effects of genetic modification on tomato	[81,82,83]
	Tomato dry-powder with water	Evaluation of the role of polyamines in growth and development in tomato	[84]
	Tomato juice/pulp	Metabolite characterization of tomato juice and pulp/Metabolite profiling of 50 different tomato cultivars	[85,86]
	St. John’s wort	Comparison of St. John’s wort extracts that have been subjected to the same standardization procedure/Prediction of pharmacological efficacy of different extracts	[87,88]
HPLC-NMR	Gingko (Gingko biloba)	Investigation of metabolomic composition of 16 commercially available Ginkgo preparations and identification of flavonoid glycosides and terpene trilactones	[89]

**Table 2 foods-10-01249-t002:** Selected literature examples of food authentication via metabolomics approaches.

Food	Study	Analytical Method	Statistical Analysis	References
Olive oil	^1^H NMR fingerprints of virgin olive oils (VOOs) from the Mediterranean basin (three harvests) were analyzed by principal component analysis (PCA), linear discriminant analysis (LDA), and partial least-squares discriminant analysis (PLS-DA) to determine their geographical origin at the national, regional, or PDO level	^1^H- and ^13^C-NMR	ANOVA PCA LDA PLS-DA	[150]
Olive oil	NMR-based metabolomics approach used to classify the extra virgin olive oils based on their geographical origin	^1^H-NMR	PLS-DA SIMCA	[151]
Honey	NMR was used as an analytical tool to confirm honey origin	^1^H-NMR	--	[152]
Honey	Proton NMR was used in conjunction with multivariate analysis techniques (PLS, PLS-LDA) to classify honey into groups by geographical origin including honey from Corsica Island and other part of Europe	^1^H-NMR	PLS PLS-LDA PLS-GP	[70]
Honey	41 honey samples from different countries were analyzed using ^1^H-NMR spectroscopy in conjunction with PCA to distinguish between polyfloral and acacia honey samples	^1^H-NMR	PCA PLS-DA	[149]
Milk	The application of attenuated total reflectance mid-infrared (MIR) microspectroscopy was evaluated as a rapid method for detection and quantification of milk adulteration. Milk samples were purchased from local grocery stores (Columbus, OH, USA) and spiked at different concentrations with whey, hydrogen peroxide, synthetic urine, urea and synthetic milk	^1^H-NMR	SIMC PLS	[153]
Beef	There is a need for new, non-invasive, rapid and reliable analytical methodologies that can easily be implemented and used for authentication of cattle production systems and the meat derived from them. Easily quantifiable markers could strengthen the current tracing methods for beef authentication. This study investigated the use of a nuclear magnetic resonance-based metabolomic approach as a tool to authenticate beef on the basis of the pre-slaughter production system	NMR	PCA PLS-DA	[154]
Cabbage	Cabbage (Brassica rapa ssp. pekinensis) is one of the most popular foods in Asia and is widely cultivated in many countries for the production of lightly fermented vegetables. In this study, metabolomic analysis was performed to distinguish two cultivars of cabbage grown in different geographical areas, Korea and China, using ^1^H-NMR spectroscopy coupled with multivariate statistical analysis. PCA showed clear discrimination between extracts of cabbage grown in Korea and China for two different cultivars (Chunmyeong and Chunjung)	NMR	PCA	[62]

## Data Availability

All data are referenced through the manuscript.

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
