# Peer review of "You Are What You Eat: Application of Metabolomics Approaches to Advance Nutrition Research"

_foods, 2021, doi:10.3390/foods10061249_

Round 1

Reviewer 1 Report

Emwas et al. described the different applications of metabolomics technologies in food science. The sections describing the food science applications part (Section 4 onwards) are well written and will serve as an important documentation for the field. However, I believe the technical parts need some re-working/writing. My specific comments are below - 

Major points - 

1. The techniques seem to be overly descriptive. I think that is not necessary as there are already several excellent technical reviews available in the literature. This is true for both NMR and MS sections. I suggest the authors make these sections concise, as much as they can.

2. The authors should consider adding a section for general sample preparation for food samples.

3. The tables should provide some quantitative context throughout the manuscript, but especially in the technical sections. For example, instead of saying low/high resolution in Table 4, the authors should provide an approximate number with the relevant unit.

Minor points - 

1. line 106: Untargeted analysis doesn't necessarily mean unbiased. Please re-structure the sentence.

2. Table 1 left panel heading "column type" should be changed to "separation technique" or chromatographic technique".

3. The authors should thoroughly check for grammar and typos as well.

Author Response

Dear Reviewer,

first of all, together with all authors I would like to thank you for your valuable revision and suggestions regarding our manuscript, to which we all agree to. Please find in the following brief descriptions of the changes we made in order to improve our paper, in particular with regard to data presentation.

Revisions are listed below:

Major points  

  1. The techniques seem to be overly descriptive. I think that is not necessary as there are already several excellent technical reviews available in the literature. This is true for both NMR and MS sections. I suggest the authors make these sections concise, as much as they can.

Reply: The section 3 (Metabolomics analytical tools) was shorten, and only the most relevant data were left in the proper order.

  1. The authors should consider adding a section for general sample preparation for food samples.

Reply: The review is already lengthy, and we prefer not to add additional sections.

  1. The tables should provide some quantitative context throughout the manuscript, but especially in the technical sections. For example, instead of saying low/high resolution in Table 4, the authors should provide an approximate number with the relevant unit.

Reply: Table 4 was cancelled, while the third section was shortened.

Minor points

  1. line 106: Untargeted analysis doesn't necessarily mean unbiased. Please re-structure the sentence.

Reply: The sentence was re-written.

  1. Table 1 left panel heading "column type" should be changed to "separation technique" or chromatographic technique".

Reply: Table 1 was cancelled, while the third section was shortened.

  1. The authors should thoroughly check for grammar and typos as well.

Reply: Grammar and typos were corrected.

Best regards

Joanna I. Lachowicz

Reviewer 2 Report

The aim of this review is to provide a background on metabolomics applied to food research. Authors explain common and basic metabolomics knowledge, as well as details on analytical techniques used in the field. This topic is of interest. However, I have many concerns that need to be addressed. My main critic is that some parts of the manuscript are long, repetitive (are 200 references really necessary?), and lacks of consistency. Additionally, some parts do not follow a clear structure in the text and/or are not accurate. For all my comments, see below:

Main issues:

1) Authors devote a big part of the manuscript (pages 4-9) to describe common knowledge and basic principles of analytical techniques. Most of these are not needed. Additionally they do not really use correct terminology to the analytical chemistry field and some basic analytical chemistry fundaments are not well explained. Thus, I would recommend authors to only focus on metabolomics applications to food research. Thus, completely restructuring/deleting section 3. There is no need to explain in such specific details what HPLC, GC or MS are. My recommendation is that authors only adhere to explaining the advantages and limitations of each technique when they think this is relevant. To illustrate my point, some examples are below:

  • Line 161, 164: LC and MS are techniques, not methods.
  • Line 161: the P from HPLC means performance, not pressure! (they correctly define this on 132). Why to introduce abbreviations over and over if they were already defined on first appearence?
  • Line 164: “HPLC separates the molecules present in the sample”.
  • Line 166: What do authors mean with fractions? This might mislead readers with fractionated HPLC.
  • Lines 168-175: This description of how HPLC works is not accurate. Depending on the polarity of the analytes these will interact differently with the stationary phase and thus will be retained (or not). The sentence that the analyte is present in the mobile phase, while not completely wrong, is not accurate. The analyte is not formerly present in the mobile phase but it “pushes” the analytes through the column. I recommend to suppress this paragraph and just redirect authors elsewhere.
  • Line 189: UPLC is the commercial term from Waters trademark. The correct name for this technique is UHPLC (Ultra High Performance Liquid Chromatography). (Oddly, authors properly define this correctly again on line 598…).
  • Line 204: what do authors mean here with vacuum? The ESI is atmospheric pressure ionization, the nebulization does not occur in a vacuum phase.
  • No need to explain every single source ion in Table 2, especially if these are not even commented afterwards.
  • Table 3 talks about packed vs capillary column but lines 237-240 talks about PLOT and WCOT.

2) Why authors decide to focus on NMR based metabolomics approaches on section 7? This is not supported by the title nor the abstract of the manuscript. Yet, authors include a non-NMR work on lines 641-659. And, additionally, although they have not mentioned capillary electrophoresis technique at all in the manuscript they include a capillary electrophoresis work here as well (lines 773-781).

3) There is no consistency with the abbreviation. Even though NMR has been mentioned tons of times, on line 596 authors again introduce this term…

Others:

Line 92 makes reference to a supplementary Table but this is not included?

Lines 142-143: NMR is not per se an easy technique to identify metabolites with. I would have strengthened more the fact that it is inherently a quantitative technique.

Lines 153-156: do authors think that these couplings are really used in practice? I think their use is really marginal and not representative.

Line 259: I don’t see why petroleum samples are something worth to mention in a manuscript concerning food research?

Line 300: right after talking about the fundaments of NMR, authors jump into nutritional metabolomics. This should be introduced differently or should have its own section.

Table 5 is not very informative as only the matrix used is included. Why don’t authors include the aim of the study there for example?

Author Response

Dear Reviewer,

first of all, together with all authors I would like to thank you for your valuable revision and suggestions regarding our manuscript, to which we all agree to. Please find in the following brief descriptions of the changes we made in order to improve our paper, in particular with regard to data presentation.

Revisions are listed below:

Main issues

  • Authors devote a big part of the manuscript (pages 4-9) to describe common knowledge and basic principles of analytical techniques. Most of these are not needed. Additionally they do not really use correct terminology to the analytical chemistry field and some basic analytical chemistry fundaments are not well explained. Thus, I would recommend authors to only focus on metabolomics applications to food research. Thus, completely restructuring/deleting section 3. There is no need to explain in such specific details what HPLC, GC or MS are. My recommendation is that authors only adhere to explaining the advantages and limitations of each technique when they think this is relevant.

Reply: The section 3 (Metabolomics analytical tools) was shorten, and only the most relevant data were left in the proper order.

To illustrate my point, some examples are below:

  • Line 161, 164: LC and MS are techniques, not methods. Reply: Has been corrected.
  • Line 161: the P from HPLC means performance, not pressure! (they correctly define this on 132). Reply: This paragraph was cancelled. Why to introduce abbreviations over and over if they were already defined on first appearence?
  • Line 164: “HPLC separates the molecules present in the sample”. Reply: This paragraph was cancelled.
  • Line 166: What do authors mean with fractions? This might mislead readers with fractionated HPLC. Reply: This paragraph was cancelled.
  • Lines 168-175: This description of how HPLC works is not accurate. Depending on the polarity of the analytes these will interact differently with the stationary phase and thus will be retained (or not). The sentence that the analyte is present in the mobile phase, while not completely wrong, is not accurate. The analyte is not formerly present in the mobile phase but it “pushes” the analytes through the column. I recommend to suppress this paragraph and just redirect authors elsewhere. Reply: This paragraph was cancelled.
  • Line 189: UPLC is the commercial term from Waters trademark. The correct name for this technique is UHPLC (Ultra High Performance Liquid Chromatography). (Oddly, authors properly define this correctly again on line 598…). Reply: Has been corrected.
  • Line 204: what do authors mean here with vacuum? The ESI is atmospheric pressure ionization, the nebulization does not occur in a vacuum phase. Reply: This paragraph was cancelled.
  • No need to explain every single source ion in Table 2, especially if these are not even commented afterwards. Reply: Table 2 was cancelled.
  • Table 3 talks about packed vs capillary column but lines 237-240 talks about PLOT and WCOT. Reply: Table 3 was cancelled.

  • Why authors decide to focus on NMR based metabolomics approaches on section 7? This is not supported by the title nor the abstract of the manuscript.

Reply: The title of section 7 was changed into: NMR in analysis of food components.

Yet, authors include a non-NMR work on lines 641-659.

Reply: This paragraph was removed.

And, additionally, although they have not mentioned capillary electrophoresis technique at all in the manuscript they include a capillary electrophoresis work here as well (lines 773-781).

Reply: This paragraph was removed.

  • There is no consistency with the abbreviation. Even though NMR has been mentioned tons of times, on line 596 authors again introduce this term.

Reply: The use of abbreviations was corrected.

Others

Line 92 makes reference to a supplementary Table but this is not included? Reply: Table S1 is included now.

Lines 142-143: NMR is not per se an easy technique to identify metabolites with. I would have strengthened more the fact that it is inherently a quantitative technique. Reply: The sentence was re-written.

Lines 153-156: do authors think that these couplings are really used in practice? I think their use is really marginal and not representative. Reply: The sentence was re-written.

Line 259: I don’t see why petroleum samples are something worth to mention in a manuscript concerning food research? Reply: This phrase was removed.

Line 300: right after talking about the fundaments of NMR, authors jump into nutritional metabolomics. This should be introduced differently or should have its own section. Reply: proper changes has been made.

Table 5 is not very informative as only the matrix used is included. Why don’t authors include the aim of the study there for example? Reply: additional column (‘Purpose of study’) was added to the table (now Table 1).

Best regards

Joanna I. Lachowicz

Round 2

Reviewer 1 Report

In the revised version of the manuscript, the authors thoroughly addressed my previous concerns.

Author Response

Dear Reviewer,

Together with all authors I would like to thank you for your valuable revision and suggestions regarding our manuscript.

Best regards

Joanna I. Lachowicz

Reviewer 2 Report

Authors have made an effort to address some of my comments and have improved the manuscript. However, some remarks I made previously still remain unaddressed, and some aspects still need polishing.

Section 3: it is not clear why authors dedicate so much time to capillary HPLC when this is hardly used in metabolomics. How many of the works they detail later on actually use this technique? Same for FT-IR (line 134). No need to mention them if they are not relevant for this field of study! Please, adhere to the techniques that are used in metabolomics and the ones that are relevant to this specific review.

Section 3 is still repetitive: in lines 152 authors say that a single analytical technique cannot be used to study the whole metabolome, yet authors repeat themselves on line 171…

Line 173: the way to solve the issue described above, i.e. a single technique cannot cover the whole metabolome, would be to use independent and complementary techniques, i.e. multi-platform set up. Meaning that one needs to use for instance LC-MS, GC-MS and NMR. As I already said in my first revision, the combinations of setups described by authors are not really in use in metabolomics.

Table 1: Ref 70. Is Ilex considered as food?

Lines 686-689: despite the fact that authors focus on NMR contributions in section 7, yet, as I said in my previous comments, they include other techniques for some reason. In this case Ref. 136 is a LC-MS, please, be consistent. Also, it should be included somewhere why authors decide to focus on NMR and not in other techniques. Same applies for Refs 154, 159, in Table 2. Additionally, if authors want to really focus solely on NMR this should be included in the abstract and conclusions of the manuscript as well.

Table 2 includes statistical models and analytical techniques in a single column called “methods”, this is not correct and must be adjusted accordingly.

Author Response

Dear Reviewer,

Together with all authors I would like to thank you for your valuable revision and suggestions regarding our manuscript, to which we all agree to. Please find in the following brief descriptions of the changes we made in order to improve our paper:

Section 3: it is not clear why authors dedicate so much time to capillary HPLC when this is hardly used in metabolomics. How many of the works they detail later on actually use this technique? Same for FT-IR (line 134). No need to mention them if they are not relevant for this field of study! Please, adhere to the techniques that are used in metabolomics and the ones that are relevant to this specific review.

Reply: Proper correction in lines 136-145 were made.

Section 3 is still repetitive: in lines 152 authors say that a single analytical technique cannot be used to study the whole metabolome, yet authors repeat themselves on line 171.

Reply: The sentence in lines 151-153 was removed.

Line 173: the way to solve the issue described above, i.e. a single technique cannot cover the whole metabolome, would be to use independent and complementary techniques, i.e. multi-platform set up. Meaning that one needs to use for instance LC-MS, GC-MS and NMR. As I already said in my first revision, the combinations of setups described by authors are not really in use in metabolomics.

The phrases in lines 171-177 were re-written accordingly.

Table 1: Ref 70. Is Ilex considered as food?

Reply: Ref 70 was cancelled.

Lines 686-689: despite the fact that authors focus on NMR contributions in section 7, yet, as I said in my previous comments, they include other techniques for some reason. In this case Ref. 136 is a LC-MS, please, be consistent.

Reply: Ref 136 was cancelled.

 Also, it should be included somewhere why authors decide to focus on NMR and not in other techniques.

Reply: The proper comments were added in lines 72, 73, 182.

Same applies for Refs 154, 159, in Table 2. Additionally, if authors want to really focus solely on NMR this should be included in the abstract and conclusions of the manuscript as well.

Reply: The references related to other (than NMR) techniques were removed from Table 2.

Table 2 includes statistical models and analytical techniques in a single column called “methods”, this is not correct and must be adjusted accordingly.

Reply: Table 2 was improved according to the suggestions.

Best regards

Joanna I. Lachowicz